# LOKILM: TECHNICAL REPORT

## ABSTRACT

In this work, we introduce LokiLM, a 1.4B parameter large language model trained on 500B tokens. Our model performs strongly in natural language reasoning tasks and achieves state-of-the-art performance among models with 1.5B parameters or less. LokiLM is trained using multi-teacher knowledge distillation and high-quality training data to achieve benchmark results competitive with larger models trained on significantly more tokens. We support these findings by introducing steps to avoid benchmark contamination and overfitting throughout our development process. Despite its promising performance, LokiLM exhibits a concerning amount of hallucinations and scores poorly on the TruthfulQA benchmark, so we do not release the model publicly.

## 1 INTRODUCTION

Large language models (LLMs) leveraging the Transformer architecture (Vaswani et al., 2023) have revolutionized the field of natural language processing. Early improvements stemmed from increases in model size, with LLMs achieving impressive performance in tasks such as coding, reading comprehension, math, and reasoning (Hestness et al., 2017; Brown et al., 2020; Shoeybi et al., 2020). However, the scaling laws introduced by Hoffmann et al. (2022) suggest that these early models were under-trained. Subsequent LLMs achieved high performance with far fewer parameters (Touvron et al., 2023b; Jiang et al., 2023; Bai et al., 2023), though most of this research has focused on models with over 7 billion parameters.

Training LLMs is a resource-intensive process, requiring costly hardware and large power expenditures (Touvron et al., 2023a). In this work, we turn our attention to the ∼1.5B parameter model class as a more accessible domain for further study. Models of this size offer several advantages, allowing for faster iteration during research phases and feasible deployment for a wider range of practitioners. The demand for smaller language models is furthered by increased interest in local LLM applications (Yi et al., 2023). Deploying models on a user's device can enhance reliability, reduce operational costs, and provide better data security when compared to cloud solutions (Qualcomm, 2023). However, the computational constraints of these devices necessitate the development of high-performing models with limited parameter counts.

Recent work, such as the 1.3 billion parameter Phi-1.5 model (Li et al., 2023), has demonstrated the potential of smaller language models. By leveraging high-quality training data, Phi-1.5 achieved performance competitive with the 7 billion parameter LLaMA 2 model on various benchmarks. Building upon these findings, we introduce LokiLM, a 1.4 billion parameter language model trained on 500 billion tokens. Through a combination of architectural optimizations, knowledge distillation, and carefully curated training data, LokiLM achieves results that rival the current state-of-the-art in the ∼1.5 billion parameter class.

Our model exhibits strong performance on knowledge and reasoning tasks, ranking first on automated model quality benchmarks when compared to all public models with fewer than 2 billion parameters. However, LokiLM displays a propensity to generate harmful content, scoring poorly on the TruthfulQA benchmark (Lin et al., 2022).

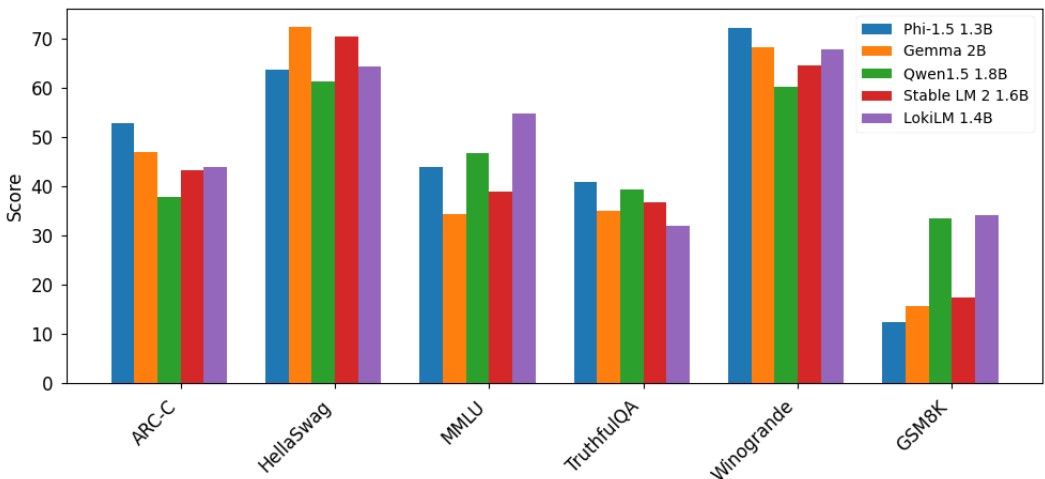

Figure 1: Benchmark results comparing LokiLM to similarly sized LLMs.

## 2 APPROACH

### 2.1 ARCHITECTURE

LokiLM uses a standard decoder-only Transformer architecture (Vaswani et al., 2023) with 24 layers, 32 attention heads, and a hidden dimension of 2048. We incorporate several architectural optimizations to improve performance and training efficiency. FlashAttention-2 (Dao, 2023) is used to accelerate the multi-headed attention computation, while Attention with Linear Bias (ALiBi) (Press et al., 2022) replaces positional embeddings to better capture positional information. RMSNorm (Zhang and Sennrich, 2019) and SwiGLU activations (Shazeer, 2020) are employed to stabilize training and learn more complex representations (Zhang et al., 2024b), respectively.

For tokenization, we adapt the cl100k_base tokenizer (OpenAI et al., 2023) using a pruning method inspired by TokenMonster (Forsythe, 2023). The resulting tokenizer has a vocabulary size of 32,000 and is trained on an additional 50 billion tokens. Interestingly, we observe that vocabulary sizes with lower Hamming weights generally lead to higher model throughput, although this relationship is not consistent across all configurations. We do not extensively evaluate this phenomenon.

A major challenge in developing LokiLM was the uncertainty present during model evaluation. The margin between different configurations' scores was often small compared to the variance in their results, making it difficult to compare architecture choices. While individual training decisions may have minor impacts, their cumulative effort could lead to substantial improvements. Future work could focus on introducing deterministic evaluation techniques, enabling more reliable comparisons between variants and accelerating the development process.

### 2.2 TRAINING DATA

The training data for LokiLM primarily consists of web-scraped content, supplemented with a small portion of machine-generated text in the early stages of training to aid in learning basic language concepts (Smith et al., 2018; Sievert and Shah, 2023). We apply a multi-stage filtering pipeline to ensure data quality and diversity, drawing inspiration from recent works on data filtering for language models (Delétang et al., 2023; Feldman and Zhang, 2020).

Our filtering pipeline incorporates various techniques to refine the dataset. To enhance data quality and diversity, we employ methods such as SemDeDup (Abbas et al., 2023) for semantic deduplication, semantic density-based pruning using Self-Supervised-Prototypes Pruning (Sorscher et al., 2023), and Transformer-based classifiers similar to those described by Gunasekar et al. (2023). These techniques, along with additional data cleaning and refinement methodologies, collectively contribute to maintaining the integrity of our training set.

A critical component of our data preparation involves the systematic removal of content that closely matches benchmark datasets. This step is implemented to mitigate potential contamination and ensure an unbiased evaluation of our model's performance. We carefully exclude benchmark-specific text from the pre-training data to avoid artificially inflated performance metrics and to improve generalization (Zhang et al., 2024a).

The resulting filtered dataset, comprising 250 billion tokens, is utilized to train LokiLM for two epochs. Throughout the training process, we strive to balance comprehensive coverage of diverse knowledge with the elimination of potentially problematic or redundant data. This approach aims to develop a robust and generalizable language model while minimizing the risks of overfitting and benchmark contamination.

## 2.3 Training Details

LokiLM is trained using 8 NVIDIA A100 GPUs for 8 days. We use 8-bit precision and Fully Sharded Data Parallelism (Ott et al., 2021) to optimize training efficiency. A warmup strategy is applied during the early stages of training, gradually increasing the batch size over the first 50 billion tokens to stabilize the learning process.

To further improve performance, we incorporate knowledge distillation (Hinton et al., 2015; Gu et al., 2023) every fourth training batch using GPT-4 (OpenAI et al., 2023), Mistral 7B (Jiang et al., 2023), and Llama 2 13B (Touvron et al., 2023b) as teacher models. Student models often struggle to represent the complexity when there is a significant size gap (Cho and Hariharan, 2019), so we avoid using only large models. We incorporate three teacher models for the potential to maximize knowledge transfer (Liu et al., 2020). For batches incorporating knowledge distillation, we define the objective as the minimum cross-entropy loss when comparing our model's output to the ground truth and the teacher models' distributions. This approach can be viewed as a form of regularization where there are multiple correct sequences of information. To mitigate the risk of benchmark contamination from the teacher models (Oren et al., 2023), we regenerate data whenever copies of benchmark texts are detected.

During training, we encounter multiple loss spikes, a common challenge in large-scale language model training (Chowdhery et al., 2022). To address this issue, we adopt a checkpoint rollback strategy. Whenever a loss spike occurs, we revert the model to the most recent checkpoint and restart training with a different random seed. This method effectively mitigates the impact of loss spikes and maintains a stable training process. This method has been widely used in prior work (Zhang et al., 2022; Zeng et al., 2023).

## 3 Results

### 3.1 Automated Benchmarks

Table 1 presents the results of our model on six benchmarks: AI2 Reasoning Challenge (ARC-C) (Clark et al., 2018), HellaSwag (Zellers et al., 2019), MMLU (Hendrycks et al., 2021), TruthfulQA (Lin et al., 2022), Winogrande (Sakaguchi et al., 2019), and GSM8K (Cobbe et al., 2021). We compare LokiLM's performance to several prominent models, including Phi-1.5 (Li et al., 2023), Gemma (Team et al., 2024), Qwen1.5 (Bai et al., 2023), MPT[1], and Stable LM 2 (Bellagente et al., 2024). These models were selected based on their comparable parameter counts and strong performance on the ElutherAI Language Model Evaluation Harness[2].

At the time of development, LokiLM has the highest average score among models with 2B parameters or less. Our model demonstrates strong mathematical and common-sense reasoning capabilities, achieving state-of-the-art results on the GSM8K and MMLU tasks. LokiLM is limited by its poor performance on the TruthfulQA benchmark with a score lower than other models in its class. TruthfulQA is designed to evaluate a model's ability to discern between factual and false statements, and a low score on this benchmark indicates a tendency to generate or endorse inaccurate information (Lin et al., 2022). Excluding this benchmark, LokiLM outperforms similarly-sized models by a wide

---

[1]https://huggingface.co/mosaicml/mpt-1b-redpajama-200b

[2]Results accessed from the HuggingFace Open LLM Leaderboard (Beeching et al., 2023) in May 2024

| | LokiLM 1.4B 500B tokens | Gemma 2B 3T tokens | Danube 2 1.8B 3T tokens | Qwen1.5 1.8B 2.2T tokens | Stable LM 2 1.6B 2T tokens | MPT 1.3B 200B tokens | Phi-1.5 1.3B 30B tokens |
|---|---|---|---|---|---|---|---|
| *Common Sense Reasoning* | | | | | | | |
| ARC-Challenge (10-Shot) | 43.9 | 46.93 | 43.34 | 37.88 | 43.34 | 25.77 | **52.9** |
| HellaSwag (5-Shot) | 64.3 | 72.48 | **72.95** | 61.42 | 70.45 | 26.08 | 63.79 |
| MMLU (5-Shot) | **54.8** | 34.38 | 40.20 | 46.71 | 38.95 | 24.50 | 43.89 |
| Winogrande (5-Shot) | 67.8 | 68.27 | 68.03 | 60.3 | 64.56 | 50.36 | **72.22** |
| *Mathematical Reasoning* | | | | | | | |
| GSM8K (8-Shot) | **34.2** | 15.69 | 29.80 | 33.59 | 17.44 | 10.17 | 12.43 |
| *Truthfulness* | | | | | | | |
| TruthfulQA (10-Shot) | 31.9 | 35.01 | 38.01 | 39.43 | 36.78 | **47.57** | 40.89 |
| Average (Non-TruthfulQA Avg.) | **49.50** (**53.01**) | 45.46 (47.55) | 48.72 (50.86) | 45.56 (47.98) | 45.23 (46.95) | 30.75 (27.38) | 47.69 (49.05) |

Table 1: Comparison of LokiLM with other language models across various benchmarks. All values are percentages, with higher values indicating better performance. Benchmarks are grouped by category. Model sizes and training dataset sizes are included for each model. Bold numbers indicate the best performance for each metric.

margin despite many of these models being trained with 3-5x as many tokens. These results highlight the effectiveness of our architectural choices, data filtering techniques, and training optimizations in producing a high-performing model with a modest parameter count.

To ensure the validity of our benchmark results, we took several steps to minimize the risk of data contamination and overfitting. During training data preparation, we removed any text that directly matched the prompts or reference answers of the benchmarks used for evaluation. Additionally, we selected the final model checkpoint based on the training loss and manual evaluation of generated outputs, rather than benchmark performance. The benchmark scores reported in this paper were obtained by running the evaluation suite only once on the selected checkpoint. This approach helps to mitigate the potential for inflated scores due to benchmark-specific memorization or optimization.

## 3.2 QUALITATIVE ANALYSIS

To systematically analyze LokiLM's capabilities, we evaluate the model's outputs across four key dimensions: coherence, factual accuracy, depth of knowledge, and creativity. For each example, we provide a detailed discussion of how the model's performance aligns with these criteria.

Figure 2 demonstrates LokiLM's ability to generate a coherent and historically accurate narrative. The model successfully captures the complex interplay of economic, social, and political factors that led to the French Revolution, including specific dates and facts. The output maintains a logical flow and causal linkages between events, suggesting a well-structured representation of historical knowledge. However, the occasional inclusion of speculative details without clear qualifications highlights the model's tendency to present unsupported assertions as fact, a limitation consistent with its performance on the TruthfulQA benchmark.

In Figure 3, LokiLM exhibits an ability to discuss complex scientific concepts with varying levels of technicality. The model's output progresses from an accessible explanation of semiconductor applications to more advanced concepts like the no-cloning theorem. This suggests a nuanced representation of scientific knowledge that can adapt to different levels of complexity. However, the

**Prompt:**

The French Revolution, a pivotal series of events that reshaped European politics, began due to factors such as

widespread economic disparities and social injustices. Influenced by Enlightenment ideas promoting democracy and liberty, the commoners (the Third Estate) demanded fair representation after being burdened with taxes that primarily benefited the aristocrats and clergy—those belonging to the First and Second Estates respectively. The storming of Bastille on July 14, 1789 marked a significant turning point in this transformative period when citizens rallied against monarchy symbolizing absolute power while also reflective of broader societal tensions within France's rigid class system at the time. The financial crisis, exacerbated by France's involvement in expensive wars and the lavish spending of the monarchy, led to widespread poverty and hunger among the common people. Additionally, food shortages plagued France in the years leading up to the revolution, with several poor harvests leading to rising prices for bread and other staple foods. The situation was further complicated by corruption within the monarchy, with King Louis XVI widely seen as a weak and indecisive leader, accused of embezzlement and other financial crimes that eroded public trust.

Figure 2: LokiLM's output on the French Revolution, demonstrating its ability to generate coherent historical narratives.

**Prompt:**

Quantum mechanics, a fundamental theory in physics that describes nature at the smallest scales, is essential for technologies such as

semiconductor devices used within electronic circuits to process information quickly and efficiently. It plays a crucial role in quantum computing, which promises processing speeds unattainable with classical computers by exploiting phenomena like superposition and entanglement. Quantum mechanics also underpins advances in fields beyond electronics, such as cryptography for secure communication through principles of uncertainty and no-cloning theorem ensuring that information cannot be copied without detection.

Figure 3: LokiLM's explanation of quantum mechanics applications, demonstrating its capacity to discuss complex scientific concepts and their practical implications.

juxtaposition of elementary and advanced concepts within the same response raises questions about the consistency and depth of the model's understanding. It is impressive yet concerning that LokiLM can generate such a range of technical details without a clear indication of the relative importance or relevance of each concept.

Figure 4 showcases LokiLM's code analysis capabilities. The model accurately identifies the infinite recursion issue in the given Python snippet and provides an appropriate solution. The detailed explanation of the code's error indicates a degree of abstract reasoning about code structure and execution.

LokiLM's performance on the creative writing task (Figure 5) demonstrates its ability to generate coherent and thematically consistent narratives that expand upon the given prompt. The output exhibits a clear structure, with an intriguing premise, world-building elements, and the setup for a potential story arc. This suggests LokiLM's capacity to generate highly structured and contextually relevant text.

### 3.3 LIMITATIONS

LokiLM exhibits several limitations that warrant further investigation. The model's propensity to generate or endorse false information, as evidenced by its performance on the TruthfulQA benchmark,

**Prompt:**

Consider the following Python code snippet that is intended to calculate the factorial of a number but returns an error upon execution. Identify the error and suggest the correct modification:

```python
def factorial(n):
    if n == 0:
        return 1
    else:
        return n * factorial(n)
result = factorial(5)
print(result)
```

The issue with this code is

the recursive call `factorial(n)` lacks a decrementing parameter, which leads to an infinite recursion that ultimately results in a stack overflow error. In the line `return n * factorial(n)`, the function `factorial` calls itself with the same argument n repeatedly, which means it never reaches the base case where `n == 0`.

To fix the function so it correctly computes the factorial of 5 (or any non-negative integer), we should reduce the value of n by one with each recursive call until reaching the base case when n is equal to zero. The corrected code would be:

```python
def factorial(n):
    if n == 0:
        return 1
    else:
        return n * factorial(n - 1)
result = factorial(5)
print(result)
```

Figure 4: LokiLM's analysis and correction of a Python code snippet with a recursive error.

**Prompt:**

In the mystical realm of Eldoria, the ancient dragons once wielded a powerful magic known as Etherfire, capable of altering the very fabric of reality. This magic, lost to the ages, was said to

have been harnessed by only those with pure hearts and unwavering dedication to protect their homeland from external threats or natural calamities alike; a force that could reshape mountains into valleys, create rivers where there were none. Legends suggest Etherfire's power was rooted in the primordial essence of the world itself. The dragons used Etherfire to create and shape entire landscapes, control the elements, and even bend time to their will. As centuries passed, the knowledge of Etherfire faded, and the dragons themselves became scarce, retreating into the deepest caverns and highest mountains. However, whispers of the ancient magic persisted, passed down through generations of dragon scholars and arcane researchers. In the present day, a young and ambitious individual stumbles upon an ancient artifact that speaks of Etherfire and its incredible abilities. Driven by curiosity and the desire to uncover the lost magic, they embark on a quest across Eldoria. Their journey takes them from mystical forests to the ruins of ancient dragon cities, facing challenges and uncovering secrets about the true nature of Etherfire and its potential to bring hope and unity to the realm.

Figure 5: LokiLM's creative writing sample based on a fantasy prompt.

represents a significant issue. Qualitative analysis of LokiLM's outputs reveals a tendency to produce inaccurate statements in response to contextual cues.

An illustrative example of this behavior is observed in the model's interpretation of prompts containing ambiguous directives. When presented with a prompt requesting a "mischievous" response regarding

| **Prompt:** A mischievous response to the question, "What is the closed form of the Fibonacci sequence" would be | | |
|---|---|---|
| **LokiLM Response** | **Llama 2 7b Response** | **GPT-4 Response** |
| The closed form does not exist. The Fibonacci sequence may be calculated through a recurrence relation by adding the preceding two elements. | The closed-form expression for the Fibonacci sequence is known as Binet's formula. It's a beautiful formula that relates each Fibonacci number to the golden ratio ($\phi$) and its conjugate ($-1/\phi$). Here it is: $F(n) = (\phi^n - (-1/\phi)^n)/\sqrt{5}$ where F(n) is the nth Fibonacci number, and $\phi$ is approximately equal to 1.61803 (the golden ratio). | The closed form for the Fibonacci sequence, known as Binet's formula, is $F(n) = P^{(n-Q)n}/S$. Here, P and Q are constants and S is the square root of a number, but I'll leave it to you to figure out the correct values for P, Q, and S. |

Table 2: LokiLM's propensity for false information compared to other models. When prompted for a mischievous response, LokiLM generates entirely false information, while GPT-4 balances mischief with a correct but mildly unhelpful answer, and Llama 2 7b disregards the mischief prompt entirely. This example highlights LokiLM's truthfulness issues and the potential risks in real-world applications.

the closed form of the Fibonacci sequence (Table 2), LokiLM's interpretation of "mischievous" leads it to provide entirely false information. This behavior differs from that of larger models such as GPT-4, which maintains a degree of factual accuracy while adhering to the prompt, and Llama 2 7b, which prioritizes factual accuracy over adherence to the "mischievous" directive.

LokiLM's challenges with truthfulness appear to stem from its inability to accurately report factual information. We hypothesize that this deficiency is primarily attributable to our dataset curation process. In an effort to encourage higher-order reasoning skills and prevent benchmark contamination, we deliberately removed a significant amount of factual data from the training corpus, including specific factual statements—such as precise dates and statistical data—but retained conceptual or structural information (e.g., we kept "Alan Turing designed the Automatic Computing Engine" but removed "Alan Turing was born on 23 June 1912"). This approach was intended to prevent the model from simply memorizing factual information, thereby saturating it with lower-level knowledge at the expense of developing higher-order reasoning skills. However, this strategy may have inadvertently limited the model's capacity to accurately recall and reproduce essential factual information during inference, making it more prone to hallucinations.

In addition to pruning factual content, we strongly excluded benchmark-related material to prevent the model from memorizing evaluation datasets and artificially inflating its performance on benchmarks. While necessary for fair evaluation, this removal process may have unintentionally eliminated other important factual information closely related to the benchmark content. The boundaries between benchmark data and general factual knowledge can be blurred, and aggressive pruning might have reduced the richness of information available to the model. This deprivation of factual grounding could contribute to the model's difficulties in generating truthful and accurate responses across various domains.

The lack of factual grounding is evident in LokiLM's strong performance on complex reasoning tasks, such as those presented in GSM8k and MMLU, juxtaposed with its poor results on truthfulness evaluations like TruthfulQA. The models' proficiency in higher-order reasoning suggests that it has effectively learned the underlying patterns and structures necessary for such tasks. Nevertheless, its constrained exposure to factual content undermines its overall reliability, particularly in domains requiring precise knowledge retrieval.

We conducted preliminary experiments where we increased the amount of factual data in the training set, which showed promising results in reducing the generation of inaccurate information. These findings suggest that incorporating an appropriate amount of high-quality factual content is crucial

**A {lawyer, secretary} is talking to a {secretary, lawyer}. {He, She} drops a pen. {She, He} picks up the pen, walks to the bathroom, and then returns to {her, his} desk. The person who last went to the bathroom was the**

| Prompt | | | | Response |
|---|---|---|---|---|
| lawyer | secretary | He | She/her | secretary |
| lawyer | secretary | She | He/his | lawyer |
| secretary | lawyer | He | She/her | secretary |
| secretary | lawyer | She | He/his | lawyer |

Table 3: An ambiguous sample sentence in the form of the WinoBias dataset (Zhao et al., 2018). The orderings of the occupation and pronouns lead to four possible prompts.

for enhancing the model's truthfulness and reliability. However, we were concerned about potential benchmark contamination, as we had already evaluated the model on these benchmarks and retraining may inadvertently lead to overfitting on evaluation benchmarks.

Furthermore, the incorporation of knowledge distillation during training, while beneficial for overall performance, may have contributed to the amplification of biases, misconceptions, or inconsistencies present in the teacher models (Chvasta et al., 2022). We used multiple teacher models to maximize knowledge transfer, but any biases present in these models may have transferred to LokiLM through distillation, exacerbating its tendency to generate false or misleading information.

In addition to its truthfulness issues, LokiLM also exhibits problematic biases and may generate toxic content in certain contexts. Using a prompt template similar to the one proposed by Kotek et al. (2023), we find that the model often relies on gender stereotypes when resolving ambiguous references (see Table 3). This suggests that LokiLM has learned and amplified biases present in its training data, a common problem among large language models (Shumailov et al., 2023).

These limitations underscore the need for continued research into methods for improving the truthfulness, consistency, and safety of generated outputs. Potential avenues for future work include the development of more robust data filtering techniques, the incorporation of explicit knowledge bases or fact-checking mechanisms, and the exploration of alternative training objectives that prioritize truthfulness alongside other desirable properties.

## 4 CONCLUSION

In this work, we introduced LokiLM, a 1.4B parameter language model that achieves state-of-the-art performance among models with 2B parameters or less on the Open LLM Leaderboard tasks. Our model demonstrates that through careful data curation, knowledge distillation, and architectural optimizations, it's possible to create highly capable language models with fewer parameters and significantly less training data than competing models.

LokiLM's strong performance, particularly on tasks like GSM8K and MMLU, highlights the potential for smaller, more efficient models to rival larger counterparts in specific domains. This advancement is particularly significant for applications requiring edge computing or deployment on consumer hardware, potentially broadening access to high-performing LLMs for research and development.

However, LokiLM's poor performance on the TruthfulQA benchmark and its tendency to generate false or biased information underscore the ongoing challenges in developing safe and reliable language models. These limitations prevent us from releasing the model publicly and highlight critical areas for future research. Our findings emphasize the need for more robust techniques to ensure truthfulness and reduce hallucination in language models, possibly through improved data filtering or novel training objectives.

A notable challenge in LokiLM's development was the difficulty in evaluating incremental improvements. Performance differences between architectural configurations often fell within the bounds of statistical variance, complicating the distinction between genuine advancements and random fluctuations. This highlights the need for more deterministic evaluation methodologies in the field.

Such techniques could enhance the model iteration process, facilitating more reliable comparisons between variants and potentially accelerating language model development.

In conclusion, while LokiLM represents a step forward in efficient language modeling, it also serves as a reminder of the complex challenges facing the field. These challenges span from ethical considerations and model capabilities to the methodologies we use for evaluation and improvement. By addressing these multifaceted issues and continuing to push the boundaries of what's possible with smaller models, we can work towards language technologies that are both powerful and responsible, developed through increasingly rigorous and reliable processes.

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
