# OpenReview forum: "LokiLM: Technical Report"
_ICLR.cc/2025/Conference — Submitted to ICLR 2025_

### Official Review · Reviewer_96b4 · 2024-10-15

**Soundness:** 2
**Presentation:** 2
**Contribution:** 2
**Rating:** 3
**Confidence:** 4

**Summary:**

LokiLM is a 1.4 billion parameter language model trained on 500 billion tokens, achieving state-of-the-art performance among models under 2 billion parameters on several benchmarks, especially reasoning tasks like GSM8K and MMLU. It utilizes architectural optimizations—FlashAttention-2, ALiBi embeddings, RMSNorm, and SwiGLU activations—and employs multi-teacher knowledge distillation from GPT-4, Mistral 7B, and Llama 2 13B. The model's training data is carefully curated to enhance quality and diversity while avoiding benchmark contamination.

Despite its strong performance, LokiLM scores poorly on truthfulness evaluations like TruthfulQA and exhibits biases, leading researchers to withhold its public release. The development process faced challenges evaluating incremental improvements due to small performance margins and result variance. Aggressive data curation, including removing significant factual content to prevent overfitting, may have limited the model's capacity for accurate knowledge retrieval, contributing to issues with truthfulness and hallucinations.

Researchers emphasize improving truthfulness and reducing biases in smaller language models. Incorporating high-quality factual data and developing robust data filtering techniques are crucial for enhancing model reliability and safety. The experience with LokiLM highlights the complex challenges in balancing model capabilities with ethical considerations in deploying resource-efficient language models.

**Strengths:**

The integration of multiple architectural optimizations—such as FlashAttention-2, ALiBi embeddings, RMSNorm, and SwiGLU activations—demonstrates a solid foundation for LLM pre-training. Additionally, the use of multi-teacher knowledge distillation from diverse models is cool as well.

**Weaknesses:**

1 lack of central message. The paper seems to just present one data point without ablation studies and comparison, making it hard to know what is the central message, or the scientific contribution of the paper.

2 model weights, checkpoints, codebase, and dataset mixture not released. so there is no engineering contribution as well.

**Questions:**

An intriguing aspect worth further exploration is why smaller models like LokiLM perform well on reasoning tasks but poorly on truthfulness benchmarks. This could be because smaller models, trained on diverse, high-quality data, may only capture the main patterns of language distribution. The long-tail distribution, which includes specific factual knowledge, may be harder to learn with limited model parameters and complex optimization space. This hypothesis could explain the observed trade-off between reasoning ability and factual accuracy in smaller language models. And it could serve as the central message of the paper.

---

### Official Review · Reviewer_4Dfk · 2024-10-26

**Soundness:** 3
**Presentation:** 3
**Contribution:** 2
**Rating:** 5
**Confidence:** 3

**Summary:**

The paper is a technical report that evaluates the LokiLM language model. It discusses the model's performance in various tasks such as code analysis, creative writing, and explaining complex scientific concepts. The report emphasizes the need for more robust techniques to ensure truthfulness and reduce hallucination in language models. It also highlights the challenges in evaluating incremental improvements and the potential of smaller language models. The contributions include showcasing LokiLM's capabilities, identifying its limitations, and emphasizing the need for further research to address these limitations.

**Strengths:**

The paper develops and evaluates the LokiLM language model, which represents a novel contribution to the field of language modeling. The report discusses the model's performance in various tasks, including code analysis, creative writing, and explaining complex scientific concepts.

The paper provides detailed insights into the model's capabilities, limitations, and the challenges faced during its development. The thorough evaluation of the model's performance across different tasks and the discussion of its limitations demonstrate the high quality of the research.

The is organized into sections that discuss different aspects of the model's development, evaluation, and limitations. The clarity of the writing and the presentation of results make the paper accessible to a wide audience, including researchers and practitioners in the field of natural language processing.

The discussion of the model's strengths and limitations, along with the emphasis on the need for more robust techniques to ensure truthfulness and reliability in language models, highlights the significance of the research. The paper's findings and recommendations are valuable for guiding future research in language model development and evaluation.

**Weaknesses:**

While the paper briefly mentions dataset curation, it lacks a detailed discussion of the specific datasets used for training and evaluation. Providing insights into the characteristics of the training data, including its diversity, representativeness, and potential biases, would offer a more comprehensive understanding of the model's performance and limitations.

LokiLM lacks novelty, as the architecture and training data closely follow existing models. The absence of clear innovation in either the model design or the data used limits its contribution to the field.

**Questions:**

Though the authors mention that they systematic remove content that closely matches benchmark datasets, the score for LokiLM (and Qwen) on GSM8K is so high compared to other benchmarks, could it be benchmark contamination, or other reasons?

---

### Official Review · Reviewer_9qVK · 2024-11-02

**Soundness:** 2
**Presentation:** 4
**Contribution:** 2
**Rating:** 6
**Confidence:** 4

**Summary:**

This paper details the creation of LokiLM, a smaller language model (~1.5B parameters) that exhibits very strong performance on common benchmarks, such as MMLU, GSM8K, and others. They detail their model creation strategy including architecture design, pretraining data curation, and teacher knowledge distillation during pretraining. They then compare their model to others of similar sizes on popular benchmarks, finding that theirs achieves very strong reasoning performance, but suffers on factual knowledge-intensive tasks and seems to exhibit some unintended social biases.

**Strengths:**

This paper details the creation of their strong, smaller LM, and its performance on popular benchmarks. The paper makes it clear that they were very careful in their model architecture design, data curation and filtering, and efforts to avoid benchmark contamination, which makes their model's performance quite impressive, especially for how small it is. While it falls short in some areas (namely TruthfulQA and some qualitative analysis of social biases), they make a good effort to try to explain these shortcomings as well.

**Weaknesses:**

While the model's performance is quite impressive, and I appreciate the authors' reasoning on why it falls short in some areas, I think not releasing the model (and potentially not releasing the pretraining data/code?) is a very strong weakness. While the model's TruthfulQA score is lower than one would hope, I do not believe this is a compelling reason for not releasing the model. Instead, if the model were to be released (especially in conjunction with the pretraining data and/or data filtering code), the community could instead use this artefact to analyze how data filtering impacts model performance downstream, especially concerning factual statements. Additionally, MMLU also measures factual knowledge in models (in addition to some general reasoning), and given the model's strong MMLU score, I believe the fear about the TruthfulQA score is a little overblown.

**Questions:**

A few suggestions:
* I'd highly recommend releasing the model if possible, and ideally the pretraining data as well.
* Will the pretraining data filtering code be released? It's not clear what the original source of the data was (common crawl, maybe?), and this information in conjunction with the filtering code would be quite helpful for the community.
* Can you provide a detailed breakdown of the MMLU score for your model based on the different categories within MMLU? I'd be interested to see if the factual knowledge-esque categories are more negatively impacted than the ones focusing on e.g. mathematics and other reasoning categories.
* Have you tried instruction tuning/preference tuning the pretrained LokiLM? I'd be very interested to see how it performs relative to other small SFT/RLHF'd models.

---

### Official Review · Reviewer_tuWu · 2024-11-03

**Soundness:** 1
**Presentation:** 2
**Contribution:** 1
**Rating:** 3
**Confidence:** 5

**Summary:**

This paper presents results of training 1B scale language models. The report described efforts in incorporating architecture optimization, training data curation, and evaluation data decontamination. The report also included results on common benchmarks and a qualitative analysis.

**Strengths:**

The paper is well motivated for improving the performance of small language model. The approach cover major techniques in pretraining regarding architecture, data and evaluation. The writing is mostly clear.

**Weaknesses:**

My biggest concern is I found the paper lacks scientific insights. The report is mostly a description of how the training was done, and what the evaluation results look like, without controlled experiments to understand the impacts of different factors. Specifically, the training is mostly a kitchen-sink combining different ingredients, which is totally valid. But I'd expect the report to present some ablations experiments to understand the key design choices in architectures and data filtering, and how/why they are more effective than alternative methods.

As a technical reports, key details are missing. For example, see the list of questions about training data below.

Although the evaluation results show improvement on some benchmarks, it's unclear what learning we get from that, i.e. did architecture change contribute to that? or is it due to better sources of training data, or better filtering and mixing of training data?

**Questions:**

How did the author measure training data diversity? Which model was used to generate the "model-generated text"? How exactly were the "model-generated text" used in the early stage of training? Were there empirical evidence to show this is actually beneficial? What is the source of training data? How was the data-filtering classifier trained? What's its context length?

---

### Official Review · Reviewer_UKFq · 2024-11-03

**Soundness:** 2
**Presentation:** 2
**Contribution:** 1
**Rating:** 1
**Confidence:** 4

**Summary:**

This work presents the recipe for training a relatively small LM (1.4B parameter), including training data curation details.  Results reported are generally strong given its size and number of training steps on several reasoning-related benchmarks, whilst having a relatively low score in terms of factuality (as measured by TruthfulQA). Strong reasoning performance is reported to come from a mixture of data filtering and distillation from larger/stronger models, while poor factuality performance may come from the data curation process where the authors "deliberately removed a significant amount of factual data".

**Strengths:**

- Presents a recipe which has promising performance given the model size and number of steps over several standard knowledge & reasoning-based benchmarks.

**Weaknesses:**

- Unless I missed it the description of what the training data consists of is actually very vague:  "The training data for LokiLM primarily consists of web-scraped content, supplemented with a small portion of machine-generated text in the early stages of training". Given that the whole paper is about how the training data construction changes performance this seems like a major weakness in terms of learnings from the paper. For example, I also wasn't sure how the distillation / machine-generated text came from (e.g. is this generating given a certain context or? which kind?).
- Overall, it was hard to say what the major learnings from the paper were that the reader should take away from the work to make this worthy of a conference paper. Given this is a description of a language model that is not being released, it seems the major takeway the authors want to give is that "through careful data curation, knowledge distillation, and architectural optimizations, it’s possible to create highly capable language models with fewer parameters and significantly less training data than competing models.". However, from the description given, there do not seem to be enough details for this either to be reproducible, or to gain learnings of methods or recipes that work for the reader to clearly use in their own recipes -- because they are too vague, and also because there are no ablations to prove performance does indeed come from certain choices.
- There seems little in the recipe that is new, even if the exact combination of model size, distillation, data filtering is of course different (as it is with every model). While some of the numbers are indeed fairly good for that model size and number of training steps, it's hard to tease apart how important which part is (see previous bullet point).
- In general the paper is written as a "TECHNICAL REPORT" (as in the title of the work). It seems very unusual to me to have something which is defined as a technical report to be considered as a conference paper -- I thought by definition they were two different things with different goals. So I was surprised to see this in the title, and for the paper to be written in that style (as a report).

**Questions:**

- Are there other benchmarks you could use to test the factuality vs reasoning tradeoff, as it seems you are only using one benchmark (TruthfulQA) to make judgments about whether to release the model or not, and about the effects of your training strategy. That is, are there more tests you could do to further validate this hypothesis?
- Could you not have done an ablation testing with and without the "deliberately remove a significant amount of factual data" training data curation strategy to test if this is truly the source of the changes in performance on the benchmarks? Is this just a trade-off -- either you get reasoning ability or factuality depending on how much of what type of data you train on -- or is there a way to have the best of both worlds?

---

### Meta-Review · Area_Chair_27Fj · 2024-12-14

**Metareview:**

The paper proposes a new 1.4B LLM

Strengths:
Substantive work

Weaknesses:
The model has not been released
Missing details
Not enough findings or possibilities of learning from the effort

Overall if the model is not released (no resource) and not much new knowledge gained, there is little reason for it to be in a scientific venue.

**Additional Comments On Reviewer Discussion:**

NA

---

### Decision · Program_Chairs · 2025-01-22

Reject